# Formation of Ag-Fe Bimetallic Nano-Species on Mordenite Depending on the Initial Ratio of Components

**DOI:** 10.3390/ma16083026

**Published:** 2023-04-11

**Authors:** Yulia Kotolevich, Evgenii Khramov, Perla Sánchez-López, Alexey Pestryakov, Yan Zubavichus, Joel Antúnez-Garcia, Vitalii Petranovskii

**Affiliations:** 1Centro de Nanociencias y Nanotecnología, Department of Nanocatalysis, Universidad Nacional Autónoma de México, Ensenada 22860, Mexicojoel.antunez@gmail.com (J.A.-G.);; 2Kurchatov Complex for Synchrotron and Neutron Studies, National Research Center “Kurchatov Institute”, Moscow 123182, Russia; 3Research School of Chemistry and Applied Biomedical Sciences, Tomsk Polytechnic University, Tomsk 634050, Russia; 4Synchrotron Radiation Facility SKIF, Boreskov Institute of Catalysis SB RAS, Koltsovo 630559, Russia

**Keywords:** mordenite, silver, iron, ion exchange, Ag/Fe ratio

## Abstract

The formation and properties of silver and iron nanoscale components in the Ag-Fe bimetallic system deposited on mordenite depend on several parameters during their preparation. Previously, it was shown that an important condition for optimizing nano-center properties in a bimetallic catalyst is to change the order of sequential deposition of components; the order “first Ag^+^, then Fe^2+^” was chosen as optimal. In this work, the influence of exact Ag/Fe atomic proportion on the system’s physicochemical properties was studied. This ratio has been confirmed to affect the stoichiometry of the reduction–oxidation processes involving Ag^+^ and Fe^2+^, as shown by XRD, DR UV-Vis, XPS, and XAFS data, while HRTEM, S_BET_ and TPD-NH_3_ show little change. However, it was found the correlation between the occurrence and amount of the Fe^3+^ ions incorporated into the zeolite’s framework and the experimentally determined catalytic activities towards the model de-NOx reaction along the series of nanomaterials elucidated in this present paper.

## 1. Introduction

A question about the nature of active centers is always a key problem in catalysis. Catalytic properties of nano-centers depend on their structure and composition influenced by factors such as particle size, effective surface charge, amount of defects, the strength of metal–support interaction, etc. [1,2,3]. A rational approach to the creation of catalytic materials is the competent choice of a metal–carrier combination and the study of an optimal number of physicochemical parameters that characterize the activity of surface nano-centers.

Zeolites have vital importance as catalysts due to their unique characteristics. First, they have the properties of acid catalysts and are also widely used as supports for active metals or reagents [4,5]. Second, application of these materials combining of micro- and mesoporosity opens new perspectives for the development of chemical technology: size and shape selectivity allows better control of reagents and products of catalytic reactions [4]. The most important characteristic is that many catalytically relevant transition metal cations can be anchored to the cation—exchange sites at the internal surface of the zeolite’s pores by straightforward wet chemistry with a specific post-synthesis treatment to afford nanomaterials with tuned chemical and electronic characteristics but preserved desired porous structure [4,6].

The fabrication of bimetallic zeolite-based materials has aroused much attention as an instrument for fine tuning the catalytic properties of such materials. Taking into account that the functional performance of these nanomaterials is governed by the local environment of deliberately introduced transition metal ions, it is of great importance to study the methods for tuning objects of the local structure through synergistic metal-support or metal–metal interactions [5,7,8,9,10]. X-ray absorption spectroscopy (XAS) was used to study highly dispersed species since it applies to non-crystalline and nanostructured materials. More specifically, XAS modification, Extended X-ray Absorption Fine Structure, or EXAFS, is a tool for the direct measurement of essential bond lengths involving the incorporated transition metal ions and their proximate coordination environment. Moreover, EXAFS can be implemented in the so-called in situ and operando modes to probe the state of the incorporated cation affected by external conditions, such as elevated temperature and reactive atmosphere [11].

Widely studied mechanisms of strong metal–support interaction are metal–metal bonding, specific morphology and structure, surface charge transfer, and surface mass transport [12]. However, metal–metal redox interactions remain unclear [3]. In our previous works, the coupled reduction–oxidation processes converting Ag^+^ and Fe^2+^ into Ag^0^ and Fe^3+^ were considered from the perspective of the formation of various iron and silver species, the state of which is influenced by the order of cation deposition [13,14,15]. As shown, the predominant fraction of the Ag species in all specimens under study remains in a cationic form, and the structural state of Ag^+^ in the mordenite structure is independent of the specific sequence of the two metal introduction; in this case, silver clusters, metallic and colloidal unstructured nanoparticles (NPs) are formed, which are in different proportions, influenced by the order of ion deposition [13]. The coordination of silver in the zeolite is complex and includes four inequivalent bond lengths formed by Ag and its atomic neighbors: O, Si, or Al [14]. For both orders of cation deposition, the Fe–O bond lengths were similar to those found in Fe_2_O_3_, containing a minor fraction of iron in the Fe^2+^ oxidation state [15].

The literature about metal–support interaction mostly describes the formation of active sites (i.e., clusters, nanoparticles, faces, vertices, edges, etc.) and collaborative effects, such as electron-donating effect or oxygen transport [16]. Besides those, there are several structural metal–support interactions; for example, the incorporation of Ag^+^ into the mordenite framework gives rise to its significant structural distortion [14]. Transition-metal-ion-substituted zeolites are believed to manifest attractive catalytic activity in addition to the proneness to further cation exchange. However, there are very few reports on the preparation of iron-bearing zeolites characterized by a larger cation exchange capacity (CEC) [17,18,19]. Based on our previous results, a mechanism was proposed, which assumes that the framework Al^3+^ gets isomorphously substituted for Fe^3+^. This process is palliated by the presence of Ag^+^. This gives rise to novel efficient catalytic sites, especially promising for the deNOx reaction. The framework-incorporated Fe^3+^ ions are additionally promoted by proximate Ag^+^ (i) and a minute fraction of Ag nanoparticles dispersed on the Fe-modified mordenite surface (ii) [20].

The purpose of this present work is thus to elucidate the effect of varying Ag/Fe ratios on the redox interaction of silver and iron and the isomorphous substitution of framework Al^3+^ for Fe^3+^. For this present study, the component deposition order “first Ag^+^ then Fe^2+^” was selected because in such a case, according to our earlier reported results, both effects are most intense.

## 2. Materials and Methods

### 2.1. Sample Preparation

Mordenite in the Na-exchanged form (NaMOR) characterized by a Si/Al atomic ratio of 6.5 was fabricated by Zeolyst Int., Conshohocken, PA, USA (Product CBV-10A). Precursors of Ag and Fe 0.03 N AgNO_3_ and FeSO_4_ aqueous solutions were taken. To suppress hydrolysis, the solution was deliberately acidified with H_2_SO_4_ to pH of 2, according to recommendations based on previous studies [21]. Both Ag^+^ and Fe^2+^ precursors were sequentially incorporated into the pristine NaMOR via a standard ion exchange protocol implying that the respective solution is maintained for 24 h at 60 °C. The zeolite powder was filtered, then rinsed and dried at 110 °C for 20 h in air after each ion-exchange treatment. All preparative steps were carried out to strictly avoid direct light illumination to maximally suppress possible photo-induced reduction of Ag species.

More specifically, a load of 2 g of NaMOR was treated with solutions containing the same total equivalent concentration (normality) of corresponding metal. The synthesis of monometallic analogs includes a single ion-exchange step with the appropriate metal following exactly the same protocol. The synthesis of bimetallic Ag and Fe-containing samples that are the focus of this present study implies two successive ion-exchange steps; where first step is ion exchange of NaMOR with silver cations, and second is with iron. However, in this case, the molar fractions of the metals to be incorporated were varied according to Table 1.

### 2.2. Characterization Methods

The chemical composition analysis of the samples under study was accomplished using inductively coupled plasma optical emission spectrometry (ICP-OES) method with a Varian VISTA MPX CCD Simultaneous spectrometer. The samples were preliminarily degassed and then dissolved in a mixture of HNO_3_ and HF, kept at 40 °C overnight. To the solution formed, a solution of H_3_BO_3_ was added at 40 °C for 5 h.

The porosity parameters were measured using the technique of nitrogen adsorption–desorption isotherms at 196 °C by Micrometrics TriStar 3000 equipment. First of all, the samples were evacuated at 300 °C for 5 h for degassing. The adsorbed N_2_ amount was recalculated to standard temperature and pressure. The specific surface area (S_BET_) of the samples was determined within the BET approach from experimental nitrogen adsorption data limiting the P/P_0_ range to 0.005–0.250. The mean effective pore size was calculated according to Barret–Joyner–Halenda (BJH) prescription based on adsorption and desorption branches of the N_2_ isotherms. The total volume of gas-accessible pores was calculated from isotherms at P/P_0_ = 0.99.

High-resolution transmission electron microscopy (HRTEM) micrographs were acquired with a Jeol JEM 2010 microscope (Tokyo, Japan) operating at an accelerating voltage of 200 kV. The zeolite powders were finely ground and suspended in isopropanol under ultrasonic assistance at RT. Afterwards, the suspension was dropped to a Lacey carbon Cu grid. To get representative statistics, no less than ten images were taken for each sample. Particle size distribution histograms were compiled based on sizing data for no less than 450–500 particles.

X-ray powder diffraction study was performed with a Panalytical X’Pert diffractometer equipped with an X-ray tube with a Cu anode operated at 45 kV and 40 mA (Cu Kα radiation, λ = 0.154 nm). Diffraction patterns were acquired over a 2θ range of 5–55° with a step of 0.02° and dwell time of 1 s per angular step. The full-profile analysis was conducted with the X’Pert HighScore Plus software Vercoin 2 (Malvern Panalytical Ltd., Malvern, UK).

DR UV-Vis (diffuse reflectance UV-Vis absorption) spectroscopy was made with a UV-Vis-NIR Cary 5000 spectrometer (Agilent Technologies, Santa Clara, CA, USA) with a diffuse reflectance unit and BaSO_4_-coated integrating sphere. Before measurements, the background signal was measured on a reference sample (Teflon). The obtained DR UV-Vis spectra measured at RT were represented as Tauc plots.

X-ray photoelectron spectroscopy (XPS) was implemented with a customized system (SPECS GmbH, Berlin, Germany) based on a PHOIBOS 150 WAL hemispherical analyzer and a non-monochromatic X-ray source Al Kα (1486.6 eV, 200 W). Pass energy was set to 50 eV, a high-intensity lens mode was selected with a scanning step of 0.1 eV. The size of the sample area being probed is 3 mm. Binding energy (BE) shifts due to charging were compensated using the signal of adventitious carbon with a C 1 s line assumed to occur at 284.5 eV. The vacuum level in the analytical chamber was not worse than 1 × 10^−8^ mbar. The nominal accuracy of BE determination was ca. ±0.1 eV.

Apparent acidity of the surface was determined in a quartz fixed bed miniature reactor operating in a continuous flow mode. The sample load was 50 mg, temperature was ramped over a range of 100–550 °C at a ramp rate of 10 °C/min. The acidity value was determined from TCD–TPD curves measured with a Chem BET Pulsar TPR/TPD apparatus (Quantachrome, Boynton Beach, FL, USA). The samples were pre-dried in a flow of He (99.99% purity) at a flow rate of 120 mL/min within a temperature range of 100–550 °C at a heating rate of 10 °C/min. The sample was then kept at 550 °C for 1 h and then cooled down to 100 °C. As a chemisorption probe, anhydrous ammonia with a purity of 99.98% supplied by Aldrich (St. Louis, MO, USA) was applied and fed at a rate of 120 mL/min for 10 min at 100 °C. The adsorbed molecules were then desorbed using a He flow for 40 min.

The transmission Ag K-edge and Fe K-edge EXAFS spectra were taken at the Structural Materials Science beamline of the Kurchatov Synchrotron Radiation Center (NRC Kurchatov Institute, Moscow, Russia). Channel-cut monochromators with 2 different crystal orientations were used: Si (220) and Si (111) for Ag K-edge and Fe K-edge, respectively. The X-ray beam intensities before and after the samples were recorded with 2 ionization chambers filled to 1 atm with pure Ar and Xe gases, respectively. Raw data processing was performed with IFEFFIT software Version ifeffit-1.2.11 package (Matthew Newville, The University of Chicago, USA) [22,23].

## 3. Results and Discussion

In this study, the charge-compensating cations balancing the negative charge of the zeolite framework are Ag^+^, Fe^2+^, and H^+^ (from H_2_SO_4_, which we use to control pH to prevent hydrolysis of ferrous sulfate solution), Fe^3+^, which is inevitably formed during the performed processes, along with the residual of Na^+^ (if any remains). The total amount of cations needed for the zero net charges of the zeolite framework is often referred to as the equilibrium ion-exchange modulus (EIEM), which should be unity. As it is impossible to estimate the content of protons by ICP-OES, in our case,
$$\mathrm{EIEM}=\frac{\frac{C_{{\mathrm{Na}}^{+}}}{1}+\frac{C_{{\mathrm{Ag}}^{+}}}{1}+\frac{C_{H^{+}}}{1}+\frac{C_{{\mathrm{Fe}}^{2+}}}{2}+\frac{C_{{\mathrm{Fe}}^{3+}}}{3}}{C_{\mathrm{Al}}}$$

It is also impossible to determine the Fe^2+^/Fe^3+^ ratio from ICP-OES data, so for all samples containing Fe, both EIEM-Fe^2+^ and EIEM-Fe^3+^ values were estimated for divalent and trivalent cations, respectively; their real EIEM of the obtained samples should have an intermediate value (in Table 2, EIEM-Fe^2+^ if *n* = 2 and EIEM–Fe^3+^ if *n* = 3).

All samples, except AgMOR, were subjected to acid treatment, which could be a reason for weak dealumination (Si/Al increases slightly, see Table 2). NaMOR has an excess content of Na^+^ in the pore space, which was explained in [13] by the possibility of insufficient washing of the commercial material and, accordingly, by the presence of residues of the NaOH used in the synthesis. Excessive Na^+^ is removed during the first step of the ion-exchange procedure; therefore, it is absent in the samples prepared for this study. For Fe-containing samples, H^+^ occupies exchangeable cationic positions, but it is impossible to evaluate its amount. If nominal and experimental Ag/Fe ratios are compared, the nominal content of silver turns out to be ~7–15 times higher than its value. The reason is the different adsorption capacities of mordenite with respect to Ag and Fe cations, which was previously discussed in [14].

The specific surface area and porosity parameters are weakly affected by the details of the ion-exchange treatment, which indicates the absence of significant destruction of the zeolite lattice. The Brønsted surface acidity of the nanomaterials under study was probed qualitatively and quantitatively using ammonia TPD over a temperature range of 100–550 °C. The respective experimental results are compiled in Table 3. Ammonia is desorbed from all samples as 2 distinct peaks: the low-temperature desorbtion observed at about 180 °C (l-peak), which is assigned to weakly acidic sites, and the high-temperature desorbtion observed at about 250 °C (h-peak), which can be attributed to strongly acidic sites. The balance between the weak and strong acidic centers for most of the studied samples does not change and is close to 30:70. In the case of FeMOR, the concentration of weakly acidic sites decreases, whereas the concentration of strongly acidic sites increases, with a ratio equal to 16:84. It is also important that for Fe-containing samples, the total acidity increases significantly when compared with NaMOR, probably due to presence of H_2_SO_4_ in the solution and the concomitant replacement of cationic sites by protons.

X-ray powder diffraction patterns of studied samples are demonstrated in Figure 1. The positions of the peaks remain approximately the same after the zeolite modification with Ag and/or Fe ions. Relative intensities variations of several peaks due to sensitivity to Fe or Ag incorporation could be explained by the destructive interference effect due to the ionic modification of the mordenite lattice. These effects were discussed in detail in our previous work [24]. The Ag metal phase (Ag 00-003-0921) with crystallite sizes (X-ray coherent scattering region) of 6 nm was observed for bimetallic samples only. The formation of Ag^0^ is explained by the occurrence of a redox reaction involving Ag^+^ and Fe^2+^, as mentioned before [13,14]. The Scherrer crystalline size (X-ray coherent scattering region) of silver (111) achieves 5.0, 7.6, and 6.2 nm for Ag3FeMOR, AgFeMOR, and 3AgFeMOR, respectively. The difference in Ag^0^ crystalline size for bimetallic samples is not significant. For a more detailed study of the silver particle sizes, the HRTEM method was applied.

In all cases, the apparent distribution of particle sizes is unimodal; the mean size of silver nanoparticles for all bimetallic samples is approximately equal to 4 nm. The main difference of AgFeMOR from other bimetallic samples (Figure 2b) is that 18% of its particles have a size of 6–21 nm, which is visible in XRD; this is up to 91% of total silver loading. However, it could be concluded that both methods confirmed the absence of particle agglomeration in the studied range of silver content. The influence of the Ag/Fe ratio in mordenite on the silver crystalline size (XRD) as well as on the silver particle size (HRTEM) is relatively negligible, and both are determined by the silver–mordenite interaction. Histograms of particle size distribution of the AgMOR sample demonstrate particles with a similar average diameter to bimetallic samples. However, the phase of metallic silver in the case of AgMOR was not found. This could mean that either the fraction of metallic particles in AgMOR is too low or the particles themselves are too small to be detected by XRD.

To get more insights into the chemical state of iron and silver in the MOR-based samples, DRS UV-Vis spectroscopy was applied (Figure 3). The studied samples demonstrate absorption lines in the regions of 200–250, 250–300, and 300–450 nm and a weak plateau at 350–700 nm. The highest intensity of absorption was manifested at 200–250 nm. However, since various neutral and charged clusters of Ag [25,26,27,28,29,30,31,32] and Fe [33,34] have adsorption in this range, the peaks of this interval cannot be assigned unambiguously. Each of the spectra of Fe-containing samples exhibits a peak at 269 nm, attributed to charge transfer within the O-Fe(III) atomic pair of FeO_4_ tetrahedral units [35,36,37]. The intensity of absorption at 269 nm decreases in a range AgFeMOR ≈ Ag3FeMOR > FeMOR > 3AgFeMOR, which corresponds to the iron content, in accordance with ICP-OES data (see Table 2). The absence of absorption at 400 and 650 nm, at which absorption occurs by larger iron oxide nanoparticles and/or polymeric species [38], confirms the effective stabilization of Fe(II) against hydrolysis thanks to adding H_2_SO_4_ to pH = 2.

A broad but weak shoulder in the absorption spectra at approximately 360 nm is distinct for all Ag-containing samples. The authors of [39] attributed it to absorption by quasi-colloidal Ag nanoparticles (with a size as small as only 1 nm) formed on the clinoptilolite crystal outer surface. An even weaker shoulder is observed at approximately 416 nm; its position is characteristic of the plasmon resonance peak of Ag nanoparticles (1–5 nm), which is easily and often observed in a system with reduced ultrafine silver [40]. Its weak intensity in bimetallic samples may be due to the action of iron ions [41,42]. Such particles have a greater size than the diameter of regular mordenite channels, so we should assume that they are located on the external surface of zeolite grains or at certain positions in mesopores of the zeolite matrix [43]. AgMOR shows wide structureless absorption in the region of 300–650 nm, which could correspond to reduced forms of silver species with a wide particle size distribution [25,40,43,44,45]; in such a case, we assume the formation of silver “blacks”, detected by HRTEM [46,47]. It should be noted that no special efforts were made to restore silver in our case. To understand the phenomena occurring in the system, the electronic properties of samples were studied.

Broad Ag 3d XPS lines (Figure 4a) were deconvolved into several non-equivalent components. This treatment shows that in all Ag-containing samples, there are three different Ag surface states (Table 4). The data interpretation of XP-spectra was taken from the literature [48,49,50,51,52]. For all Ag-containing samples, the signal with the largest contribution appeared at a BE around 369 eV assigned to the smallest Ag clusters less than 2 nm. There are more such clusters in AgMOR than in other samples, which allows them to be classified as silver blacks in accordance with the results of DR UV-Vis (see Figure 3). BE of the main Ag band, in this case of AgFeMOR, is shifted down by 0.2 eV, corresponding to Ag^0^. This could be explained by the formation of large (9–21 nm) silver nanoparticles (Figure 2b) and larger silver crystallites (Figure 1), which agrees well with both diffraction and electron microscopy data. The smaller Ag 3d_3/2_ peaks, observed at BE around 366 eV and ~367–368 eV, correspond to silver cations and the Ag–support interaction [53], respectively. The contribution of these two picks is minimal for AgMOR (20%). For 3 bimetallic samples, the sum of the contributions of these 2 electronic states is 32–34%; a negligible influence of the Ag/Fe ratio corresponds to a small difference in the iron content between Ag3FeMOR, AgFeMOR, and 3AgFeMOR (0.9, 0.9, and 0.6 at%, respectively, see Table 2). However, from the XPS data, we conclude that the presence of Fe enhances the interaction of silver with the surface of mordenite, which indirectly confirms the suppression of the agglomeration of silver atoms and the dormancy of a large number of particles exhibiting a plasmon resonance peak.

Since XANES spectra (Figure 5a) are almost coincident for all samples and differ from bulk references with a simple structure, i.e., Ag foil, Ag_2_O, etc., we suggest that virtually the entire amount of Ag is embedded into the mordenite structure with the formation of a very similar local structure for all samples containing Ag. In addition, the presence of Ag^0^ does not have any significant influence on the shape of XANES spectra; thus, the XANES method indicates that the predominant fraction of Ag in the bulk of the samples occurs in the form of Ag^+^-MOR, while some Ag^+^-MOR was identified on the surface by XPS (see Table 4).

The EXAFS K-edge data of Ag (Figure 5b) reveal a set of distinct peaks in a range from 1 Å to 2.4 Å, with general shapes almost the same for all samples and a weaker and more diffuse peak at 2.6 Å, which is clearly seen only for bimetallic samples. It could be interpreted as the first coordination sphere in metal Ag. The emergence of the Ag-Ag component corresponds well to XRD results, assuming the emergence of a bulk nanocrystalline Ag phase. Maxima at smaller distances (1–2 Å) are observed for all samples with slight variations in intensity. Previously, using theoretical references, it was demonstrated that these short-distance peaks arise from the interference of several components in EXAFS spectra [14]. It needs to be recognized that these peaks are not a structural pattern but an artifact of the rendering of the spectra. This phenomenon, called ‘interference crest’, sometimes occurs when two or more interatomic distances (e.g., Ag-O) are close but not equal and is explained in detail in our previous work [14].

To simulate the local atomic environment of Ag, we hypothetically set four Ag-Si scattering paths, suggesting that Ag substitutes Si or Al atom in the mordenite framework, although this is physically impossible. In this case, several Ag-Si (or Ag-Al) components arise at similar but not precisely identical distances, thus inducing the interference crest effect. Still, the two major maxima of the interference crest pattern are appreciably similar to those encountered for oxide Ag species (see, e.g., Ag_2_O reference shown in Figure 5b). Keeping this in mind, we also constructed a structural model with four slightly different Ag-O scattering paths, envisaging that silver ions interact with O atoms in the pores of the matrix. To minimize the parameter number, the Debye–Waller factors for the whole set of Ag-O and Ag-Si shells are equated to each other. A single Ag-Ag shell is additionally included in the model to describe Ag nanoparticles. Table 5 are shown the best-fit results. From the variation of coordination numbers, we assume that an additional O atom should be located near the Ag site in the AgMOR structure. Notwithstanding, the introduction of an extra Ag-Ag gives no improvement to the model, yielding either higher R factors or zero coordination numbers for the Ag-Ag shell. This could mean that silver occurs in AgMOR as atomically dispersed species in the oxidation state of Ag^+^.

For bimetallic samples, the Ag-Ag coordination number changes along with the intensity of the Ag Bragg peak observed by XRD. Despite the model of interference crest being uncertain due to a large number of parameters, some changes in Ag local structure in bimetallic samples are also observed. A set of Ag-O distances is similar for all samples; in accordance with XANES data, the nearest oxygen surroundings of the Ag atom remain unchanged regardless of the presence of Fe and deposition order. However, the set of Ag-Si/Al distances changes in the presence of large amounts of Fe, and the 2 shortest Ag-Si/Al distances become equal: 2.92 Å for AgFeMOR and 2.913 Å for Ag3FeMOR. This observation will probably be useful in further studies of Ag metal centers in mordenite.

Fe K-edge XANES spectra are demonstrated in Figure 6a. The entire set of Fe-containing samples reveals a pattern characteristic of Fe_2_O_3_. A simple model was applied: two nonequivalent Fe-O distances with equal Debye factors. The fit results are shown in Table 6. All bimetallic samples have coordination numbers close to 2 + 2 (similar to Fe_2_O_3_), but for AgFeMOR and Ag3FeMOR, the first coordination number is slightly decreased, and for 3AgFeMOR, the second coordination number is slightly increased. We suggest that iron is present in oxidized forms of both ionic states Fe^2+^ and Fe^3+^, but the distribution between these states in 3AgFeMOR is different from two other bimetallic samples. In the case of FeMOR, the coordination number corresponding to closer Fe-O distance is reduced to almost zero, which indicates that the coordination type in this sample is substantially different from Ag-doped samples. For all three bimetallic samples, XANES spectra are similar and have a shape typical for a metal incorporated into zeolite [54,55,56,57]. Thus, it can be suggested that in all bimetallic samples, Fe is incorporated into the mordenite framework. Since the coordination number by EXAFS does not lead to a change in coordination geometry by XANES, the difference between 1.9 and 2.7 is not significant for the structure. An additional oxygen atom (if any) for Ag3FeMOR can be a ligand: a water molecule, an OH group, etc.

The main Fe-O peak in EXAFS Fourier transforms for all modified mordenite is very similar to that in oxide references. However, it is of note that the Fe-O component is characterized by an asymmetric shape. Furthermore, it can be simulated by a superposition of two Fe-O components. The best-fit parameters for all samples indicate that the effective coordination numbers are 4, or 2 + 2 (2 pairs of O atoms at 2 different distances), while XANES data demonstrate that the coordination of Fe is octahedral. Previously, a structural model of AgFeMOR was approved by Mössbauer spectroscopy [20], EXAFS [15] and micro-Raman spectroscopy [14], which presumes the isomorphous substitution of Al^3+^ in the mordenite framework for tetrahedrally coordinated Fe^3+^, which is promoted by the presence of Ag^+^ ions proposed. Moreover, it was shown that the occurrence and apparent concentration of the framework Fe^3+^ ion in various samples correlate with catalytic properties in de-NOx reaction [15]. As can be seen from Figure 6a and Table 6, for Ag3FeMOR and AgFeMOR XAS spectra, coordination numbers and interatomic distances coincide. Therefore, increasing Fe concentration in those samples has no influence on Fe local structure.

In our previous work [14], the possible exchange mechanism of Ag^+^ and Fe^2+^ with NaMOR was considered. Due to the importance of these phenomena for catalytic properties, the ion exchange on the inter-channel surface of mordenite was considered in detail. Voids of mordenite were considered, including 8-member rings (green) and 12-member rings. In Figure 7, arrows and their colors indicate the channel in which the exchange of the corresponding cation occurs; Roman numerals indicate the possible amount of each cation exchange site. For the same valency of Ag^+^ and Na^+^ cations, Figure 8a shows that one-to-one ion exchange can occur in three ways: (a) exchange into 12-MR channels (position I, yellow color), (b) exchange of one or (c) two Na^+^ cations for Ag^+^ (positions II and III, respectively; green color). Meanwhile, the 12-MR channels have the largest dimensions, Ag^+^ cations will diffuse predominantly through this channel, and consequently, their exchange will occur preferentially in this channel. According to Table 2, the ion exchange of Na^+^ for Ag^+^ in the AgMOR sample is not 100%, indicating that there are still (but in a much smaller proportion than in starting NaMOR) sites with Na^+^ cations available for exchange. The exchange of Fe^2+^ with AgMOR zeolite is possible in 8-MR only. It was confirmed that without the prior presence of Ag^+^ cations, the number of 8-MR channels in which Fe^2+^ can hold relatively more. Thus, redox metal–metal interaction takes place in 8-MR:Ag^+^ MOR + Fe^2+^ → Fe^3 +^ MOR + Ag^0^↓.

As can be seen from Figure 7, 8-MR may be compensated only with charge 2+. To explain the compensation of the higher charge, such as Fe^3+^, on the interchannel mordenite surface, DFT calculations were applied.

In this work, it was considered the mordenite model with a Si/Al = 7 ratio (6Na^+^[Al_6_Si_42_O_96_]^6−^), recently reported in [54]. This value is close to our experimental value (see Table 2). Starting from that model, each of the six sodium atoms was separately exchanged for one Fe atom. After optimizing six different configurations, the one with the lowest energy was determined. Optimization of different configurations was carried out through DFT calculations using the same parameterization as in [58]. Figure 8a shows the minimum energy position of an Fe atom in mordenite (indicated by the ellipse with a dashed line). The unit cell has been extended a little further (the structure inside the black box) to show that the iron atom is lodged in the wall of one of the main channels. Figure 8b shows two different orientations of the local environment where the Fe atom is located. On the one hand, it is observed that the Fe atom is hosted in the vicinity of a six-membered ring (6-MR) with a single aluminum atom. On the other hand, it is observed that the iron atom is almost coplanar to the oxygen atoms, giving rise to binuclear Fe(II). Previous theoretical/experimental results for the BEA zeolite [19] have shown that iron is hosted mono-atomically on the 6-MR, similar to the configuration in Figure 8b. According to that study, the results indicate that this type of configuration may explain the nature of the catalytic process of converting methane to methanol in the Fe-zeolite. It is important to consider positions where Fe^3+^ is located in the mordenite framework around the channels as possible active centers [19]. Because, in our case, Fe (III) is a product of side processes, such as hydrolysis and side redox interactions, the amount of these centers is difficult to estimate.

## 4. Conclusions

A detailed discussion of the role of the order of sequential deposition of cations in the Ag-Fe bimetallic system on the mordenite surface and its effect on the AgFeMOR de-NOx catalysts led to the need to study the effect of one more parameter on the properties of this system. In this present work, for the previously chosen optimal deposition order “first Ag^+^, then Fe^2+^”, the cation ratio effect on the formation of materials and their physicochemical properties was studied. According to the results of XPS and XANES, it was confirmed that most of the silver in the mass of all silver-containing samples is present in the cationic form of Ag^+^-MOR. The Ag/Fe ratio affects the stoichiometry of the redox reaction between Ag^+^ and Fe^2+^, which is confirmed by the results of XRD, DR UV-Vis, XPS, and XAFS. Changes in the Ag/Fe ratio show the largest amount of reduced silver particles in the case of Ag/Fe = 1:1. In cases of 1:3 or 3:1, silver clusters or colloidal nanoparticles contributed more. The effect of the Ag/Fe ratio on the textural properties of mordenite and the size of silver particles is relatively insignificant.

Analysis of the data shows that the metal–metal redox interaction occurs in 8-MR mordenite, where the exchangeable silver cation is reduced by the Fe^2+^ ion and leaves the ion-exchange center due to the loss of positive charge. To compensate for the charge, Fe^3+^ is placed monatomically on 6-MR, which is described in the literature as an effective active site for catalytic reactions. This effect should be called metal–metal–carrier interaction.

The isomorphic substitution of framework Al^3+^ for Fe^3+^, as a structural metal–carrier interaction, was studied by the XAFS method; it was found that an increase in the Fe concentration in these samples does not affect the local structure of Fe. This confirms that we are dealing with a metal–metal–carrier interaction, since, in accordance with the previously proposed mechanism [20], the process of isomorphic substitution of framework Al^3+^ for Fe^3+^ is facilitated by Ag^+^.

## Figures and Tables

**Figure 1 materials-16-03026-f001:**
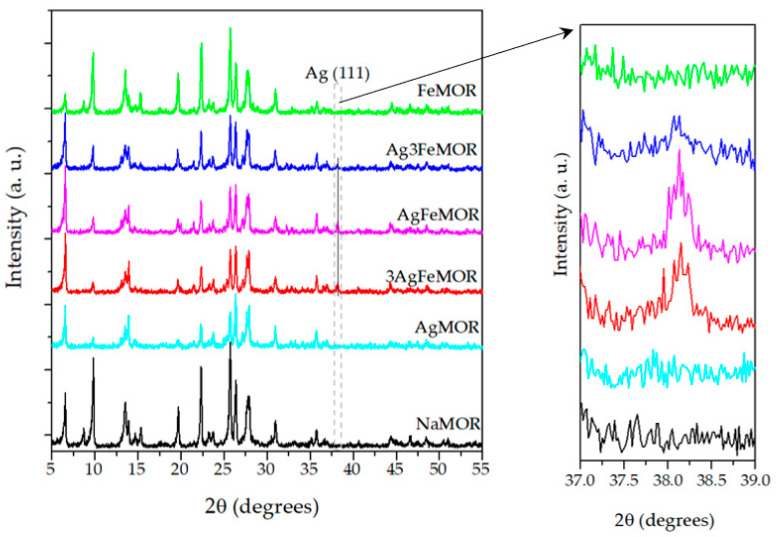
XRD patterns of studied samples.

**Figure 2 materials-16-03026-f002:**
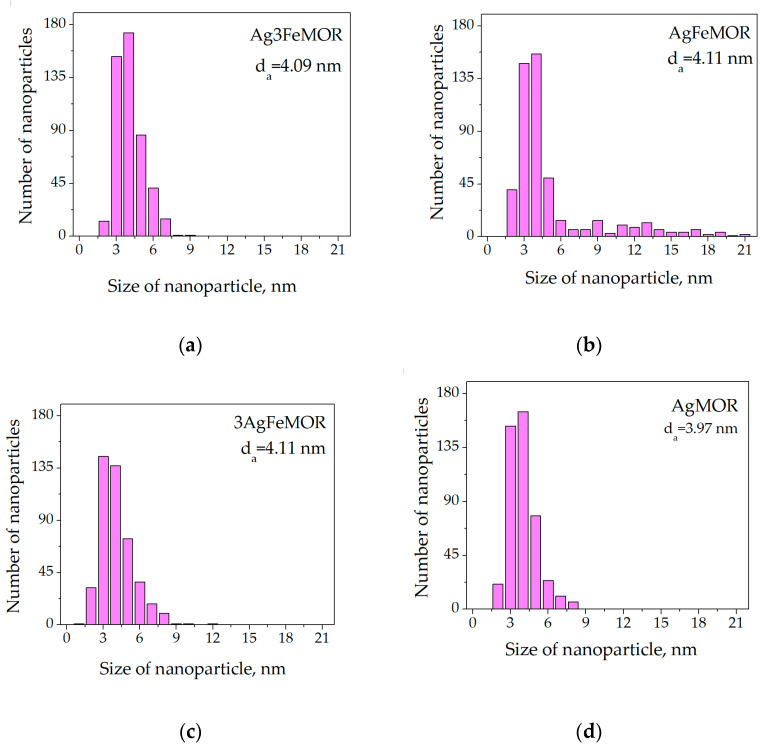
Ag particle size distribution (d_a_ is a number-averaged value of particle diameter). (**a**) Ag3FeMOR, (**b**) AgFeMOR, (**c**) 3AgFeMOR, (**d**) AgMOR.

**Figure 3 materials-16-03026-f003:**
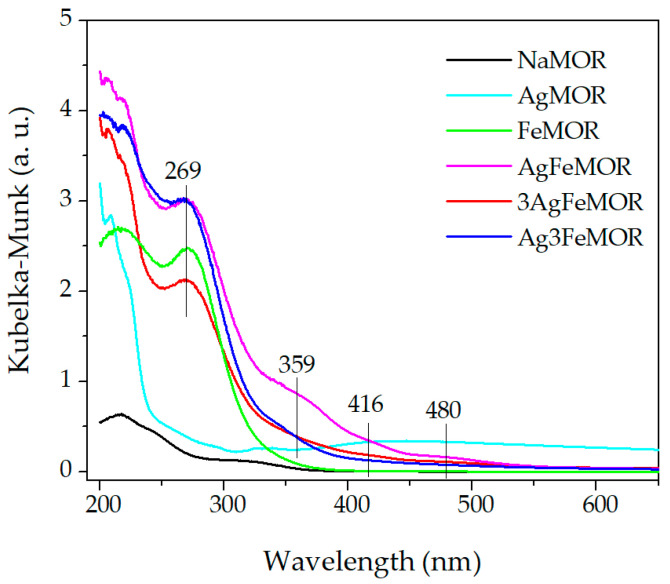
DR UV-Vis spectra of pristine support, one and two metal-modified zeolites.

**Figure 4 materials-16-03026-f004:**
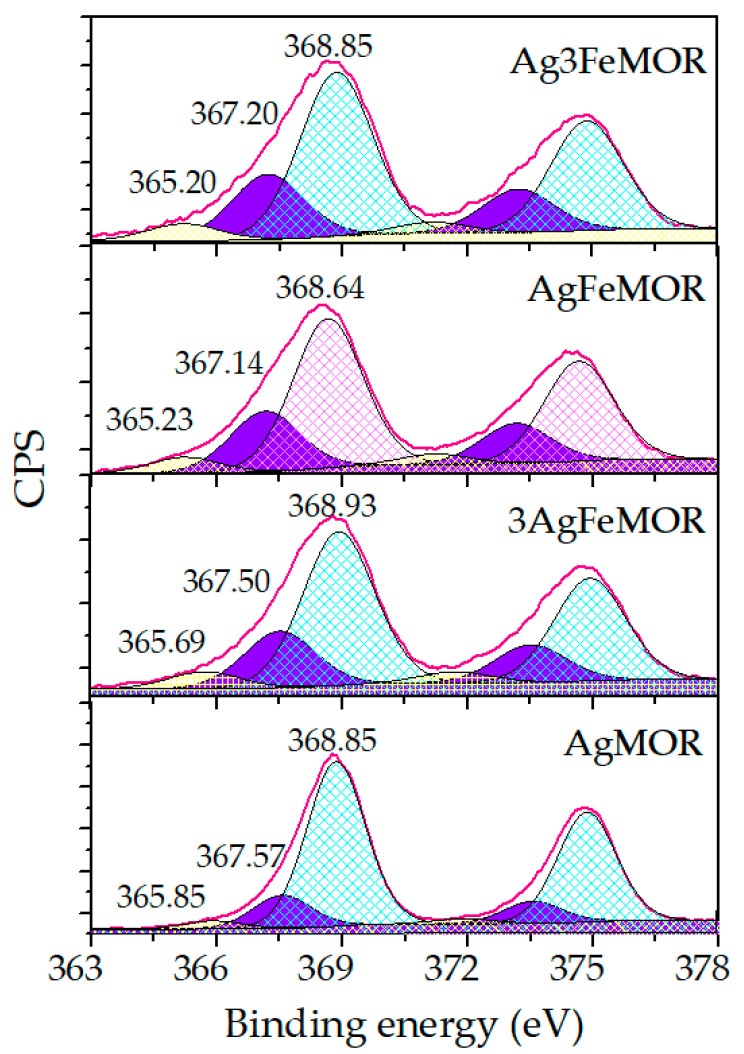
Ag 3d_5/2_ photoelectron spectra for studied samples.

**Figure 5 materials-16-03026-f005:**
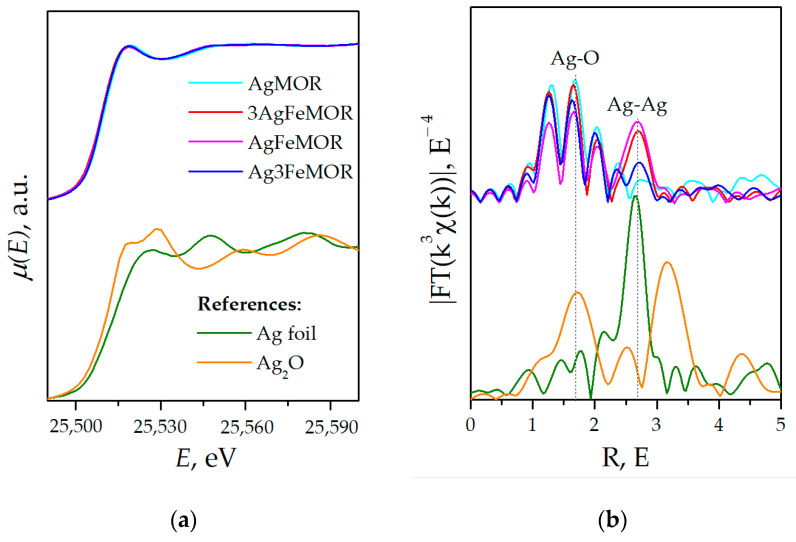
Ag K-edge XANES spectra (**a**) and EXAFS Fourier transforms (k = 0…5 Å^−1^) (**b**).

**Figure 6 materials-16-03026-f006:**
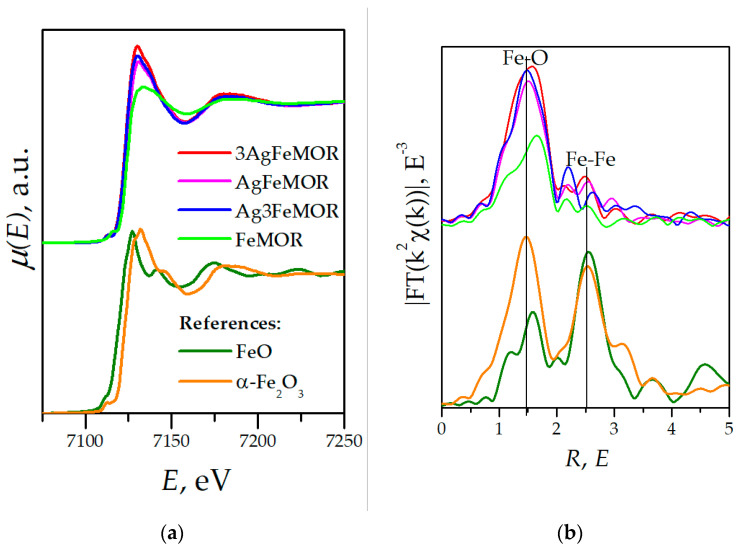
Fe K-edge XANE-spectra (normalized X-ray fluorescence spectra) (**a**) and EXAFS data (Fourier transformed taken in the K-range of 0 to 5 Å^−1^) (**b**).

**Figure 7 materials-16-03026-f007:**
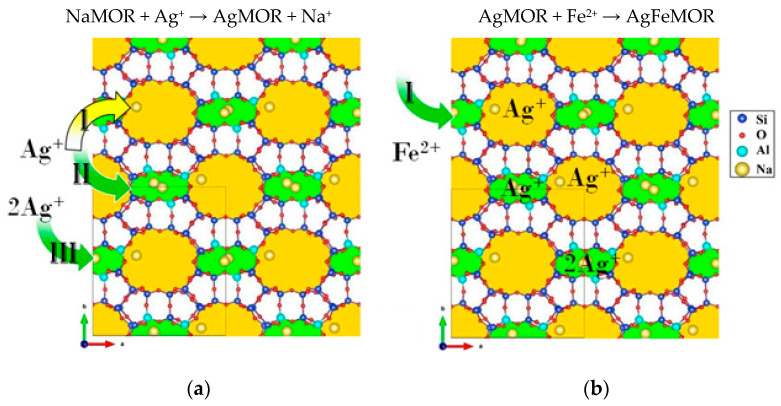
Sketch of distribution processes of Ag^+^ and Fe^2+^ cations in mordenite channels. Formation of (**a**) AgMOR and (**b**) AgFeMOR [14], where I, II and III are possible ion exchange positions.

**Figure 8 materials-16-03026-f008:**
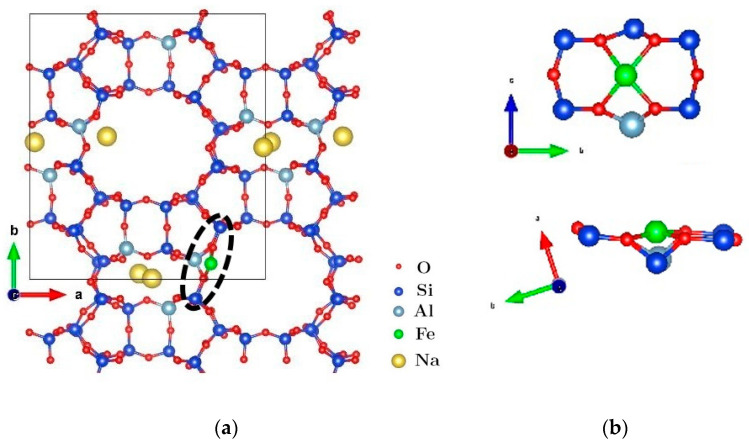
(**a**) Atomic structure for mordenite with Si/Al = 7 ratio, where the black box delimits the unit cell; (**b**) 2 different orientations of the local structure identified by the ellipse with a dashed line in (**a**).

**Table 1 materials-16-03026-t001:** Chemical compositions of the samples under study according to the synthesis.

Sample	Ion Exchange, First Step				Ion Exchange, Second Step			
	Solution, 0.03 N	V, mL	The Accessible Limit of Metal Content, Atomic %	pH	Solution, 0.03 N	V, mL	The Accessible Limit of Metal Content, Atomic %	pH
FeMOR	FeSO_4_	254	3.2	2	-	-	-	-
Ag3FeMOR	AgNO_3_	64	6.4	4–5	FeSO_4_	190	3.2	2
AgFeMOR	AgNO_3_	127	6.4	4–5	FeSO_4_	127	3.2	2
3AgFeMOR	AgNO_3_	190	6.4	4–5	FeSO_4_	64	3.2	2
AgMOR	AgNO_3_	254	6.4	4–5	-	-	-	-

**Table 2 materials-16-03026-t002:** ICP-OES results for the series of modified mordenite.

Sample	Atomic %							Ag/Fe		EIEM	
	Si	Al	O	Ag	Fe	Na	Si:Al	Nomin.	Experim.	EIEM-Fe^2+^	EIEM-Fe^3+^
NaMOR	48.9	7.5	33.9	0	0	9.7	6.5	-		1.29	
FeMOR	44.0	6.4	45.4	0	0.8	3.4	6.8	-		0.78	0.91
Ag3FeMOR	44.2	6.5	38.0	2.3	0.9	1.3	6.8	1:3	15:3	0.83	0.97
AgFeMOR	39.7	5.9	49.7	3.1	0.9	0.7	6.7	1:1	7:1	0.95	1.10
3AgFeMOR	40.6	6.0	41.8	3.7	0.6	0.6	6.7	3:1	12:1	0.92	1.01
AgMOR	38.1	5.8	49.9	4.6	0	1.6	6.5	-		1.07	

**Table 3 materials-16-03026-t003:** Porosity characteristics and acidity of the surface of the samples.

Sample	S_BET_, m^2^/g	Porosity			Acidity, %		Total Acidity, μmol/g
		Diameter, Å	V_total_, cm^3^∙g^−1^	V_micro_, cm^3^∙g^−1^	L-Peak	H-Peak	
NaMOR	243.3	22.8	0.19	0.16	32	68	1359
AgMOR	237.2	23.2	0.19	0.14	33	67	1662
3AgFeMOR	235.5	23.8	0.22	0.16	28	72	2072
AgFeMOR	251.5	23.7	0.20	0.15	29	71	2264
Ag3FeMOR	264.8	28.2	0.22	0.16	31	69	1941
FeMOR	240.6	23.3	0.23	0.17	16	84	2970

**Table 4 materials-16-03026-t004:** Ag 3d_5/2_ Binding Energy (eV) of the bands deconvoluted from Ag 3d spectra.

Samples	Ag Ions	Ag-Support Interaction	Ag^0^	Ag^0^ < 2 nm
	366.2 eV	367.4–368 eV	368.0–368.2 eV	≥369 eV
Ag3FeMOR	365.20—(7%)	367.20—(26%)	-	368.85—(67%)
AgFeMOR	365.23—(7%)	367.14—(27%)	368.64—(66%)	
3AgFeMOR	365.69—(7%)	367.50—(25%)	-	368.93—(68%)
AgMOR	365.85—(4%)	367.57—(16%)	-	368.85—(80%)

**Table 5 materials-16-03026-t005:** Best-fit values of local structure parameters from Ag K-edge EXAFS: coordination numbers N, Interatomic distances N, and Debye–Waller factors σ^2^ (R-range is from 1.2 Å to 3 Å, k-range is from 3 Å^−1^ to 15 Å^−1^, k_w_ = 3).

Sample	Scattering Path	N	R, Å	σ^2^, Å^2^	R-Factor, %
AgMOR	Ag-O	1.4	1.91	0.0001	3.1
		1.4	2.05		
		1.4	2.12		
		1.4	2.24		
	Ag-(Si or Al)	1.3	2.82	0.0020	
		1.3	2.96		
		1.3	3.02		
		1.3	3.15		
3AgFeMOR	Ag-O	1.0	1.93	0.0001	3.2
		1.0	2.08		
		1.0	2.13		
		1.0	2.26		
	Ag-(Si or Al)	1.0	2.92	0.0020	
		1.0	3.03		
		1.0	3.11		
		1.0	3.23		
	Ag-Ag	0.9	2.87	0.0073	
AgFeMOR	Ag-O	1.0	1.90	0.0001	2.3
		1.0	2.04		
		1.0	2.10		
		1.0	2.22		
	Ag-(Si or Al)	1.0	2.92	0.0028	
		1.0	2.92		
		1.0	3.12		
		1.0	3.21		
	Ag-Ag	1.6	2.82	0.0059	
Ag3FeMOR	Ag-O	1.0	1.91	0.0003	3.1
		1.0	2.05		
		1.0	2.12		
		1.0	2.25		
	Ag-(Si or Al)	1	2.913	0.0049	
			2.913		
			3.06		
			3.17		
	Ag-Ag	0.8	2.80	0.0076	

**Table 6 materials-16-03026-t006:** Best-fit values of local-structure parameters from Fe K-edge EXAFS (R-range 1.2 Å to 2.1 Å, k-range 2 Å^−1^ to 12 Å^−1^, k_w_ = 3) Scattering path-Fe-O, other notations are the same as in Table 6.

Sample	N	R, Å	σ^2^, Å^2^	R-Factor, %
FeMOR	0.6	1.85	0.0061	1.7
	1.9	2.06	0.0061	
Ag3FeMOR	1.5	1.94	0.0015	1.3
	1.9	2.10	0.0015	
AgFeMOR	1.6	1.94	0.0049	0.9
	2.0	2.10	0.0049	
3AgFeMOR	1.4	1.91	0.0044	0.3
	2.7	2.07	0.0044	

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
