# Peer review of "Formation of Ag-Fe Bimetallic Nano-Species on Mordenite Depending on the Initial Ratio of Components"

_materials, 2023, doi:10.3390/ma16083026_

Round 1
Reviewer 1 Report
It is well shown in this paper that the properties of silver and iron nanoscale components in the Ag-Fe bi-metallic system deposited on mordenite depend on several parameters applied during their preparation. As it is shown, the properties of nano-centers in a bimetallic catalyst change with the order of sequential deposition of components. When the order of introduction of silver and iron cations is chosen as first Ag+ and then Fe2+ such preparation is optimal. In this work, the authors studied the influence of exact Ag:Fe atomic proportion on the physicochemical properties of the system under study. As shown by XRD, DR UV-Vis, XPS, XAFS data this ratio affect the stoichiometry of the reduction-oxidation processes involving Ag+ and Fe2+. The isomorphic substitution of framework Al3+ for Fe3+ has been studied by the XAFS method. It is well demonstrated that an increase in the Fe concentration in these samples does not affect the local structure of Fe. In addition, the process of isomorphic substitution of framework Al3+ for Fe3+ is facilitated by Ag+. It is well prepared manuscript with very interesting results obtained using complementary physicochemical methods useful for very good description of nature and state of transition atoms in the mordenite zeolite. So, in my opinion this paper could be accepted in the present form in Catalysts journal.
Author Response
Please, find the attached file

Reviewer 2 Report
In this manuscript, authors explored the effect of ratio of Ag and Fe salts to the formation of Ag-Fe bimetallic nano-species on MOR zeolite. Some questions need be solved before accepting this manuscript.
Question 1: typos and subscribes/superscribes need to be corrected, such as Ag0, Fe3+, H3BO3.
Question 2: the formate of all figures should be consistent, including captures, legends and fonts, especially, Figure 1.
Question 3: The curve fitting of Ag K-edge is confused. Why the coordination number of Ag-O is only one value 1.4 if four scattering paths were selected. Each path should correspond to a CN value even if it is about 0. If it is 0, it means this path is not a real path. From Figure 5b, it seems there are 2 Ag-O paths, which is possible in Ag compounds. Same issues were found in other paths such as Ag-Ag and Ag-Si/Al.
Question 4: in Figure 7, the colors of Ag+ Fe2+ should be distinct and labeled to be clearer.
Question 5: "For all three bimetallic samples, XANES spectra are similar and have a shape typical for a metal incorporated into zeolite". This statement in line 373 need a citation or reference.
Author Response
Please, find the attached file

Reviewer 3 Report
In this manuscript, the authors studied the influence of exact Ag:Fe atomic proportion on the physicochemical properties of the system under study. This research is significant. Some important conclusions were obtained. I recomened that this manuscript can be accepted after minor revision.
Some comments were pionted out:
1. In XRD result, the author declared that the particles in AgMOR might not be metallic. However, XPS and XANES results revealed that metallic Ag was presented in catalysts. The relevant presentation is not strict.
2. At line 373, the elebration that “For all three bimetallic samples, XANES spectra are similar and have a shape typical for a metal incorporated into zeolite” needs be demonstrated by some literatures.
3. At the line 419, 12-MR should be marked (yellow ?)
4. The authors should carefully check and amend the format and language errors in the article.
Author Response
Please, find the attached file

Reviewer 4 Report
Comments and suggestions are reported in the attached file

Author Response
Please, find the attached file
